# Simulation of Soil Water Dynamics in a Black Locust Plantation on the Loess Plateau, Western Shanxi Province, China

**Yuting Li [1], Yang Yu [1,2,3], Ruoxiu Sun [1,2] , Mingshuang Shen [1] and Jianjun Zhang [1,2,3,\*]**

[1] College of Soil and Water Conservation, Beijing Forestry University, Beijing 100083, China; lyt0719@bjfu.edu.cn (Y.L.); theodoreyy@gmail.com (Y.Y.); sunruoxiu@bjfu.edu.cn (R.S.); smssci@126.com (M.S.)

[2] Jixian National Forest Ecosystem Research Network Station, CNERN, Beijing Forestry University, Beijing 100083, China

[3] Key Laboratory of Soil and Water Conservation & Desertification Combating, State Forestry and Grassland Administration, Beijing Forestry University, Beijing 100083, China

\* Correspondence: zhangjianjun@bjfu.edu.cn

**Abstract:** Soil moisture plays an important role in vegetation restoration and ecosystem rehabilitation in fragile regions. Therefore, understanding the soil water dynamics and water budget in soil is a key target for vegetation restoration and watershed management. In this study, to quantitatively estimate the water budget of the GFGP forests in a dry year and a wet year and to explore the recharge in deep profiles, the vertical and temporal soil moisture variations in a black locust (*Robinia pseudoacacia*) plantation were simulated under typical rainfall events and two-year cycles in a loess area between April 2014 and March 2016. We calibrated and tested the HYDRUS-1D (Salinity Laboratory of the USDA, California, USA) model using the data collected during in situ field observations. The model's performance was satisfactory, the $R^2$, Nash efficiency coefficient (*NSE*), root mean square error (*RMSE*), and mean absolute error (*MAE*) were 0.82, 0.80, 0.021, and 0.030, respectively. For the four rainfall events of 9.1 mm, 25 mm, 71.1 mm, and 123.6 mm, the infiltration amounts were 8.1 mm, 19.3 mm, 65.2 mm, and 95.3 mm, respectively. Moreover, the maximum infiltration depths were 30 cm, 100 cm, 160 cm, and >200 cm, respectively. Additionally, in the two-year model cycles, the upward average water flux was 1.4 mm/d and the downward water flux was 1.69 mm/d in the first-year cycle; the upward average annual water flux was 1.0 mm/d and the downward water flux was 1.1 mm/d in the second-year cycle. The annual water consumption amounts in the two-year cycles were 524.6 mm and 374.2 mm, and the annual replenishment amounts were 616.8 mm and 401 mm. The amounts of percolation that recharged the deep soil were only 28.1 mm and 2.04 mm. A lower annual rainfall would cause a water deficit in the deep soil, which was not conducive to the growth of *Robinia pseudoacacia* vegetation. To ensure the high-quality sustainable development of the forest land, it is suggested to adjust the stand density in a timely manner and to implement horizontal terraces to increase the infiltration and supply of precipitation. Our study provides an improved understanding of the soil water movement in *Robinia pseudoacacia* plantations and a simulated temporal moisture variation under different time scales. The results of our study provide a feasible approach for the sustainable management of *Robinia pseudoacacia* plantations during vegetation restoration.

**Keywords:** Loess Plateau; soil water movement; HYDRUS-1D simulation; Grain-for-Green Project; black locust forests

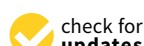



## 1. Introduction

As part of the Earth's hydrosphere, soil water is an important element in the terrestrial hydrological cycle. Soil water plays an important role in hydrological processes and vegetation restoration, which is connected with the conversion between surface water and groundwater [1]. Soil water resources affect terrestrial vegetation restoration and are one of the key indicators used to evaluate regional ecological environments [2]. Soil water shortages are one of the most severe global problems in water-limited ecosystems [3].

The Loess Plateau in China contains deep and concentrated loess with a thickness of about 350 m, and these large reserves of soil provide support for vegetation growth [4]. It is also one of the areas with the most serious soil erosion and environmental problems in China [5,6]. Due to the low rainfall and high evaporation rate in the area, water resources are very scarce. The shortage of water resources has caused variety of ecological problems, such as vegetation degradation and soil desiccation. These problems have seriously affected the sustainable development of vegetation restoration on the Loess Plateau. This has become a key issue that needs to be taken seriously. Although the Grain-for-Green Project (GFGP) implemented in 1999 has played a pivotal role in the restoration and reconstruction of vegetation in this region, progress on the Loess Plateau has been restricted by the low and concentrated annual precipitation (559.6 mm), which imposes tremendous pressure on vegetation reconstruction in this area [7]. Therefore, determining whether the soil moisture in the Loess Plateau can satisfy the vegetation growth in this area has become a research hotspot in academia and an urgent scientific problem that needs to be solved. Gaining a scientific understanding of the amount of water infiltration in the soil and the process of soil water movement in the loess area is the key to solving this problem. The soil water movement in the loess region is basically unsaturated soil flow, but it is difficult to accurately describe the water movement in the unsaturated soil because of the complexity of the soil's pore structure. This complex characteristic is manifested in the poor agglomeration and strong dispersion of the topsoil and the high density of the deep soil (200 cm). Therefore, exploring the tools and methods applicable to studying unsaturated soil water movement in the loess area is a prerequisite for determining the threshold of the ecological use of water in forest areas and a reasonable configuration of vegetation [8–10].

The saturated flow and unsaturated flow exhibit differences due to differences in their driving forces and hydraulic conductivities. The driving force of the saturated flow is the gravitational potential and the pressure potential. The pores in saturated soil are all filled with water, and the hydraulic conductivity is constant. However, unsaturated flow is affected by gravitational and matrix potentials, and parts of the pores in the soil are filled with water, which is a function of the soil water content. Unsaturated flow is the main mode state of water flow under natural conditions, so it is also the focus of this research. The method based on the physical principles used to simulate the soil water infiltration and redistribution processes is an effective way of exploring soil water movement in unsaturated zones [11]. Early studies widely used the Horton model [12], the Philip model [13], the Smith model [14], the Kostialiv-Lewis model [15], and other empirical models to describe soil water infiltration processes. Problems such as the vague physical meanings and high characteristic parameter requirements of empirical models, however, have limited their accurate description of soil water dynamics. Numerical simulation methods have become a hot topic in recent years because they can effectively simulate unsaturated soil moisture movement based on a small number of monitoring results. Several scholars have used numerical models to study soil water movement in sites with different soil types and scales, such as the Soil Water Atmosphere Plant (SWAP) model [16], the Soil and Water Assessment Tool (SWAT) model [17], the Daisy model [18], and the MODFLOW model [19]. Such numerical models, however, also suffer from limitations such as relatively fixed research scales and difficult-to-determine parameters, making it difficult to extend applications to unsaturated soil water movement.

The Hydrus-1D model based on Richard's equation has flexible input and output interfaces, integrates parameter optimization, supports the simulation of constant (constant water content or constant flux) and non-constant (variable pressure head, free drainage, deep drainage) boundary conditions, and is suitable for simulating one-dimensional vertical water movement in unsaturated soils [20]. Due to its flexible boundary conditions, the Hydrus-1D model can preferably simulate the soil-plant-atmosphere hydrological cycle, mainly including the effective rainfall in the forest, the root water absorption, the soil water stock, and the percolation [20]. At present, the model has been widely used in research on farmland irrigation, and water and salt transport, pollutant transport on the Loess

Plateau. Li et al. researched the soil moisture in deep profiles in the highlands [21]; Yu et al. analyzed soil and salt transportation under different irrigation regimes in the loess area [22]. However, few studies have been conducted on the soil water movement in the forest land established by the GFGP on the Loess Plateau. In this article, the Hydrus-1D model is used to simulate the soil water movement in the forest land on the Loess Plateau. Currently, clarifying the water balance of the plant-soil and the response mechanism of the soil water to different rainfall conditions is a key issue in formulating policies for further vegetation allocation, which the vegetation can only rely on rainfall for replenishment. The specific objectives of this study are (1) to estimate the water balance composition of the GFGP forests in a dry year and a wet year, including evapotranspiration, soil water stock and percolation; and (2) to explore whether water can be recharged to the deep soil in dry years.

As the pioneer afforestation tree for the GFGP, black locust (*Robinia pseudoacacia*) has a good soil and water conservation function and is an ideal tree species for loess areas. Many of the studies on black locust plantations have focused on carbon sequestration [23], transpiration activity [24], and productivity [25]. Few studies, however, have focused on the long-term location monitoring and simulation of soil moisture in *Robinia pseudoacacia* plantations. Therefore, in this study, we used the Hydrus-1D model to simulate the movement of water in an unsaturated soil profile in a black locust forest based on continuous soil water data observations for many years. In addition, we also investigated the redistribution of the infiltrated water in the soil profile and the response of the soil water to precipitation in order to provide a basis for optimizing the utilization of water resources and the allocation of vegetation in the loess areas.

## 2. Materials and Methods

### 2.1. Study Site and Experimental Design

The study area is located in the National Forest Ecosystem Field Scientific Observatory Station in Ji County, Shanxi Province, China. The experimental site is situated in the Caijiachuan watershed (110°39′45″–110°47′45″ E, 36°14′27″–36°18′23″ N) on the Loess Plateau. The topographic features are a typical loess gully and hilly area, with altitudes of 900 to 1589 m a.s.l. The area of the watershed is 38 km$^2$, the length of the watershed is 14 km, and the total length of the main stream and tributary stream is 51.49 km. The average drainage density is 1.35 km/km$^2$. Subject to a temperate continental climate, the annual mean temperature is 10 °C the annual sunshine hours are 2536.8 h, and the frost-free period lasts for 172 days. The annual mean rainfall is 579 mm, with the precipitation from June to September accounting for 67%, and the annual average water surface evaporation is 1732 mm (observed using Large Evaporation Panels). The main soil types are cinnamon soil with loess parent material [26]. The predominant afforestation species are *Robinia pseudoacacia*, *Pinus tabulaeformis*, and *Platycladus orientalis*; and the forest coverage in the study area is 39.8%.

We selected a *Robinia pseudoacacia* forest planted in 1991 as the long-term observation plot in the Caijiachuan watershed. We selected three 10 m × 10 m plots for the experimental sites. In addition, three soil sampling points and one root sampling point were set in each plot for soil profile investigation and root survey. The geographical location and sampling method of the sample plot are shown in Figure 1, and the basic information for the sample plots is given in Table 1.

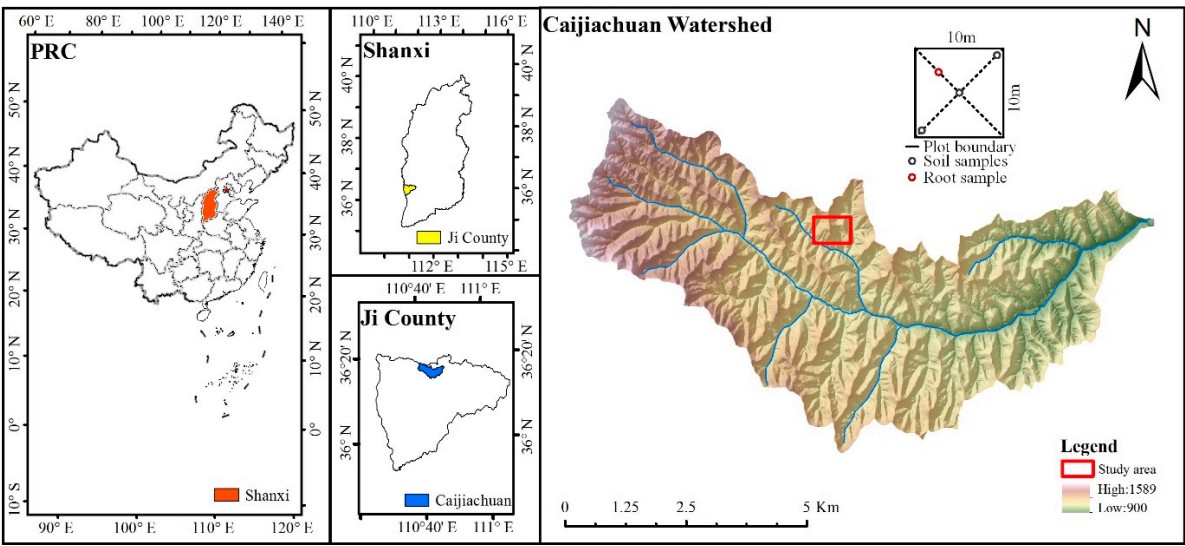

**Figure 1.** Geographic locations of the study area and sampling site. The black empty circles are the soil bulk density and soil particle composition sampling points. The red empty circles are the root density sampling points.

**Table 1.** The elemental information for the sample plots.

| Tree Species | Age (a) | Height (m) | DBH [a] (cm) | Stand Density [b] | Coverage (%) | Litter Thickness (cm) | Floor Vegetation |
|---|---|---|---|---|---|---|---|
| *Robinia pseudoacacia* | 24 | 8.05 ± 0.46 | 21.0 ± 0.11 | 1667 | 53.2 | 1.5 | *Duchesnea indica* |
| | 24 | 9.09 ± 0.17 | 22.4 ± 0.09 | 1333 | 55 | 3.0 | *Phragmites australis* |
| | 24 | 7.94 ± 0.20 | 22.7 ± 0.19 | 1333 | 53 | 2.8 | *Torilis scabra* |

[a]: Diameter at breast height. [b]: The unit of stand density is the number of plants per hectare. Values are means ± SE (standard error).

### 2.2. Soil Moisture and Meteorological Monitoring

To monitor the soil water dynamics in the locust tree plot, we embedded an EnviroS-MART (Sentek Pty. Ltd., Stepney, SA, Australia) soil moisture monitoring systems (Sentek Inc, Stepney, SA, Australia) and measured the volumetric soil moisture within a depth of 200 cm below the surface. This system consisted of a PVC pipe, sliding rail, measuring probes (installed on the sliding rail), data collector, and accumulator. The measuring probes were set in the soil at an interval of 10 cm (between 0 cm and 100 cm) and at an interval of 20 cm (between 100 cm and 200 cm). The sensors operated on the frequency-domain reflectometry principle and measured the volumetric water content ranging from oven dryness to the saturation point with a resolution of 0.1%. We collected real-time soil volumetric moisture monitoring data every 30 min and stored the data in a CR200 data collector.

We placed a HOBO rain gauge (OneSet Inc., Bourne, MA, USA) in the open space outside the *Robinia pseudoacacia* forest observation site to measure the precipitation. The rain gauge was connected to a HOBO pendant event data logger, which recorded 0.226 mm per tip. We calculated the daily precipitation between 2014 and 2016 from the event data in the logger. We obtained meteorological factors, including the maximum temperature, minimum temperature, average temperature, solar radiation, and wind speed and direction from the automatic weather station at the National Field Research Station (OneSet Inc.), which recorded data at 30-min intervals.

### 2.3. Soil Hydraulic Parameters

The soil profile was excavated to a depth of 200 cm at the three soil sampling points in the *Robinia pseudoacacia* forest. A total of 15 undisturbed soil samples were collected at intervals of 10 cm between 0 cm and 100 cm and at intervals of 20 cm between 100 cm and 200 cm. We used sample rings (250 cm$^3$) to measure the hydraulic conductivity and other soil physical properties in the laboratory, with three replicates from each layer. We used

the Ku-pF apparatus measurement system (UGT Inc., Bavaria, Germany) in the laboratory to determine the hydraulic conductivity and the pF water characteristic curve [27]. We conducted the soil particle-size analysis using a Laser Scattering Particle Size Distribution Analyzer (Microtrac Inc., Largo, FL, USA). The dry bulk density was calculated from the dry soil weight and the volume of the sampling rings (100 cm$^3$).

The hydraulic conductivity of the soil samples can be calculated using DARCY's equation assuming quasi-stationary flow conditions. This assumes that hydraulic gradient of each sample in the sample ring as constant over the length of the sample. The gradient is formed from the matrix potential (tensiometer measurement) and the gravitational potential. The hydraulic conductivity was calculated as follows:

$$V_z = k \times \frac{\Delta \varphi}{\Delta z} (DARCY), \tag{1}$$

$$\frac{\Delta \varphi}{\Delta z} = \frac{\psi_t - \psi_b}{\Delta z}, \\ Vm = \frac{1}{2}(vt - vb) = \frac{\Delta V}{2A \times \Delta t}, \tag{2}$$

$$k = \frac{\Delta V}{2A \times \Delta t} \times \frac{\Delta z}{\psi_t - \psi_b - \Delta h}, \tag{3}$$

where $V_z$ is the flow velocity of the water movement; $k$ is the hydraulic conductivity; $\varphi$ is the hydraulic potential; $z$ is the location coordinates (upward was set as positive); $\psi_t$ is the tension of the top tensiometer(the actual pressure in the unsaturated state); $\psi_b$ is the tension of the bottom tensiometer; $V_m$ is the flow velocity in the sample center; $\Delta V$ is the water volume that evaporates over the time; $\Delta t$ is the measurement interval of an individual sample; $A$ is the cross-section of the soil sample ring; $\Delta h$ is the height difference between the tension meters (3 cm); and $\Delta z$ is the distance between the tensiometers in the sample ring (3 cm).

### 2.4. Root Measurements

We measured each of the black locust trees in the three experimental plots. We selected a representative wood with average tree height and crown width as the root sampling point by removing the litter around the standard wood and excavating the soil profile (north-south orientation) with a depth of 200 cm and a length of 250 cm beside the standard wood. We set 10 root sampling points at an interval of 20 cm and collected all of the roots distributed in the 0–200 cm soil profiles. The root sample was placed in a net bag, and the soil attached to the roots was rinsed off with clean water. We measured the length and biomass of the roots in each of the sampling points. The root distribution characteristics were mainly determined for use in Equation (10).

### 2.5. Model Simulation

#### 2.5.1. HYDRUS-1D Modeling

The Hydrus-1D model is based on Richard's equations and uses the Galerkin linear finite element method to spatially discretize the soil profile while allowing for time-variable boundary conditions [28]. The model uses the implicit difference method for the time discretization, and the governing equation is as follows

$$\frac{\partial \theta(h,t)}{\partial t} = \frac{\partial}{\partial z}\left(K(h)\left(\frac{\partial h}{\partial z} + 1\right)\right) - S(h), \tag{4}$$

where $\theta$ is the volumetric water content (cm$^3$/cm$^3$); $h$ is the matric potential (cm); $t$ is the time; $K$ is the hydraulic conductivity function; $z$ is the spatial coordinate with the $z$ axis oriented downwards; and $S$ is the water absorption rate of the root system.

The van Genuchten-Mualem model [29] describes the variation of in $K(\theta)$ with soil water content:

$$\theta(h) = \begin{cases} \theta_r + \dfrac{\theta_s - \theta_r}{[1 + |\alpha h|^n]^m} & h < 0 \\ \theta_s & h \geq 0 \end{cases} \tag{5}$$

$$K(\theta) = K_s \left( \frac{\theta - \theta_r}{\theta_s - \theta_r} \right)^l \left[ 1 - \left( 1 - \left( \frac{\theta - \theta_r}{\theta_s - \theta_r} \right)^{n/(n-1)} \right)^{1 - 1/n} \right]^2 \tag{6}$$

where $\theta r$ and $\theta s$ are the residual and saturated water contents, respectively; $h$ is the pressure head; $\alpha$, $n$, and $l$ are empirical parameters; and $m = 1 - 1/n$ (dimensionless).

### 2.5.2. Initial Conditions and Boundary Conditions

In this study, the initial conditions are:

$$h(z, 0) = h_0(z), \tag{7}$$

where $h_0$ is the initial value of the pressure head in the soil profile (cm). It is assumed that the surface is the initial boundary of the recharge from precipitation.

There was no conspicuous surface runoff during the study period. The precipitation was the only water input in the model. The boundary conditions were the atmospheric conditions at the soil surface. We set the lower boundary condition for all of the cases in this study to be free drainage, which was suitable for the situation, that is, the water table was far below the bottom boundary of the soil column, and there was no groundwater recharge to the root growth zone. The upper boundary conditions were defined by the evaporation and precipitation. The potential evapotranspiration was partitioned into soil surface potential evaporation and potential transpiration from plants. We estimated the potential transpiration and evaporation using the Hargreaves equation [30] and the daily values from the weather measurements (maximum temperature, minimum temperature, and solar radiation) obtained from the meteorological stations. The boundary conditions can be described as follows:

$$-K \left( \frac{\partial h}{\partial z} + 1 \right) = 0, z = L \tag{8}$$

where $L$ is the depth coordinate of the soil surface and is equal to 200 cm at the maximum depth at which the soil dry bulk density and particle size distribution were measured and analyzed.

### 2.5.3. Root Water Absorption

We adopted Feddes' function [31] to simulate the water absorption process of the black locust root system:

$$S(z, t) = b'(z) \alpha(h) T_p \tag{9}$$

where $b'(z)$ is the relative root distribution function (dimensionless) and $\alpha(h)$ is a dimensionless water stress function. $T_p$ is the potential transpiration rate. To optimize the root water absorption, $\alpha(h)$ was set to 1 during the calibration period. The measured root data were used mainly in the $b'(z)$ function. The $b'(z)$ function [32] is as follows:

$$b'(z) = \begin{cases} \dfrac{1.6667}{L_r} & z < 0.2 L_r \\ \dfrac{2.0833}{L_r} \left( 1 - \dfrac{z}{L_r} \right) & 0.2 L_r \leq z \leq L_r \\ 0 & z > L_r \end{cases} \tag{10}$$

where $L_r$ is the root length (cm).

### 2.5.4. Spatial and Temporal Discretization

The 0–200 cm simulated soil profile was divided into 15 layers according to the soil properties. The soil profile was divided into 100 units at equal intervals of 1 cm. Correspondingly, the soil profile has 201 nodes and 6 observation points (20 cm, 50 cm, 80 cm, 120 cm, 160 cm, 200 cm). The simulated period ranges from 1 April 2014 to 31 March 2016, i.e., 731 days. Time discretization was used in the simulation, and the interval of the time discretization was gradually adjusted according to the number of iterations of the convergence. The lower and upper optimal iteration ranges were 3 and 7, respectively; and the lower and the upper time step multiplication factor were 1.3 and 0.7, respectively. If the number of iterations needed to reach reached convergence at any time in a specific period exceeded the preset maximum value (generally range 10–50), then the iteration was terminated, and the period length was changed to $\Delta t/3$ to repeat the iteration. The initial time interval was set to 0.0001 d, the maximum time step was set to 5 d.

### 2.5.5. Calibration and Validation

We divided the soil profiles into 15 layers, and the saturated water contents $\theta_s$ were calculated using the pF soil water characteristic curve measured for the soil samples in the laboratory. We estimated $\theta_r$, $K_s$, and the empirical shape parameters $n$ and $\alpha$ using the Rosetta Dynamically Linked Library (Salinity Laboratory of the USDA, California, USA) based on the neural network embedded in HYDRUS-1D from the data for the mechanical soil composition, soil bulk density, field water holding rate (the soil moisture content at $-33$ kPa), and saturated water content. Among them, $\theta_r$, Ks, $n$, and $\alpha$ are fitted parameters, and the soil's mechanical composition (sand, silt, and clay), $\theta_s$, $\rho_b$, and field water holding rate are fixed parameters with specific values derived from measurements.

The initial values of the fitted parameters for each layer were estimated using the Neural Network Prediction embedded in the model, and then, the parameters were fitted by fitting the observations measured for this layer in the experimental plot. The parameters of the van Genuchten-Mualem Equation were finally determined. The optimization results are presented in Table 2.

**Table 2.** The mechanical soil composition and calibrated hydraulic parameters used in the simulation of the different soil layers.

| Depth | Sand | Silt | Clay | | $\rho_b$ | van Genuchten-Mualem Equation Parameter | | | | |
|---|---|---|---|---|---|---|---|---|---|---|
| (cm) | (%) | (%) | (%) | Soil Texture | (g/cm$^3$) | $\theta_r$ * | $\theta_s$ | $\alpha$ * | $n$ * | $K_s$ * |
| | | | | | | (cm$^3$/cm$^3$) | (cm$^3$/cm$^3$) | (cm$^{-1}$) | | (cm/d) |
| 0–10 | 84.97 | 14.57 | 0.46 | Sandy loam | 1.29 | 0.06 | 0.41 | 0.008 | 1.9 | 133.4 |
| 10–20 | 83.12 | 16.32 | 0.56 | Sandy loam | 1.31 | 0.06 | 0.43 | 0.007 | 1.8 | 121 |
| 20–30 | 82.09 | 16.06 | 1.85 | Sandy loam | 1.34 | 0.07 | 0.51 | 0.0079 | 2 | 82.9 |
| 30–40 | 77.47 | 19.25 | 3.28 | Sandy loam | 1.35 | 0.08 | 0.5 | 0.008 | 2.2 | 52.5 |
| 40–50 | 78.64 | 19.22 | 2.14 | Sandy loam | 1.34 | 0.04 | 0.43 | 0.006 | 2.1 | 43.7 |
| 50–60 | 82.22 | 17.64 | 0.14 | Sandy loam | 1.37 | 0.08 | 0.47 | 0.007 | 2.49 | 40.7 |
| 60–70 | 83.39 | 16.61 | 0 | Sandy loam | 1.35 | 0.05 | 0.34 | 0.0039 | 2.35 | 19.5 |
| 70–80 | 81.52 | 18.48 | 0 | Sandy loam | 1.39 | 0.045 | 0.36 | 0.0054 | 2.12 | 11 |
| 80–90 | 82.91 | 17.09 | 0 | Sandy loam | 1.42 | 0.02 | 0.33 | 0.0055 | 2.06 | 8.5 |
| 90–100 | 79.43 | 17.14 | 3.43 | Sandy loam | 1.41 | 0.05 | 0.34 | 0.006 | 1.98 | 7 |
| 100–120 | 54.85 | 37.82 | 7.33 | Loam | 1.43 | 0.03 | 0.33 | 0.0041 | 2.33 | 8.1 |
| 120–140 | 53.98 | 35.22 | 10.8 | Loam | 1.45 | 0.04 | 0.3 | 0.006 | 1.93 | 9.1 |
| 140–160 | 53.17 | 34.76 | 12.07 | Loam | 1.49 | 0.03 | 0.29 | 0.00408 | 2.48 | 9.7 |
| 160–180 | 50.95 | 31.46 | 17.59 | Clay loam | 1.47 | 0.03 | 0.3 | 0.00628 | 1.78 | 8.6 |
| 180–200 | 51.17 | 33.26 | 16.57 | Clay loam | 1.49 | 0.04 | 0.3 | 0.00496 | 2.37 | 8.0 |

Note: Bulk density ($\rho_b$); residual soil water content ($\theta_r$); saturated soil water content ($\theta_s$); calibrated van Genuchten-Mualem model parameters ($\alpha$, $n$); and saturated hydraulic conductivity ($K_s$). * represents the parameters that needed to be fitted.

### 2.5.6. Assessment of the Goodness of Fit

We compared observed field measurements with the results of the HYDRUS-1D simulations using the Nash-Sutcliffe model efficiency (*NSE*), mean absolute error (*MAE*), and root mean square error (*RMSE*). Each of these values was calculated as follows:

$$NSE = 1 - \frac{\sum\limits_{i=1}^{N} (Oi - Pi)^2}{\sum\limits_{i=1}^{N} (Oi - \overline{O})^2} \tag{11}$$

$$MAE = \frac{1}{N} \sum\limits_{i=1}^{N} |O_i - P_i| \tag{12}$$

$$RMSE = \sqrt{\frac{\sum\limits_{i=1}^{N} (O_i - P_i)^2}{N - 1}} \tag{13}$$

where $O_i$ and $P_i$ are the observed and model-simulated values in the units of the particular variable; $\overline{O}$ is the measured average value for the soil layer; and $N$ is the number of observations.

## 3. Results

### 3.1. Hydro-Meteorological Conditions

An overview of the hydroclimatic conditions across the study is given in Figure 2, which presents the daily distribution of the evapotranspiration, precipitation, and mean temperature from 1 April 2014 to 31 March 2016. The two years of continuous observations revealed the seasonal characteristics of the precipitation and meteorological parameters. The average precipitation in the selected catchments during this period was 559.6 mm, and the seasonal rainfall mainly occurred from July to September, accounting for 59.4% of the total. The precipitation from 1 April 2014 to 31 March 2015 was 652 mm, and the precipitation from 1 April 2015 to 31 March 2016 was 455 mm. The two different precipitation years represented different hydrological years in the loess region, namely, the wet year and the dry year. The average annual actual evapotranspiration was 524.5 mm, and the evapotranspiration from July to September was 173.3 mm, accounting for 33% of the total. The mean air temperature during the selected period was 10.74 °C, ranging from −14.2 to 28.7 °C, and the evapotranspiration was consistent with the trend in air temperature.

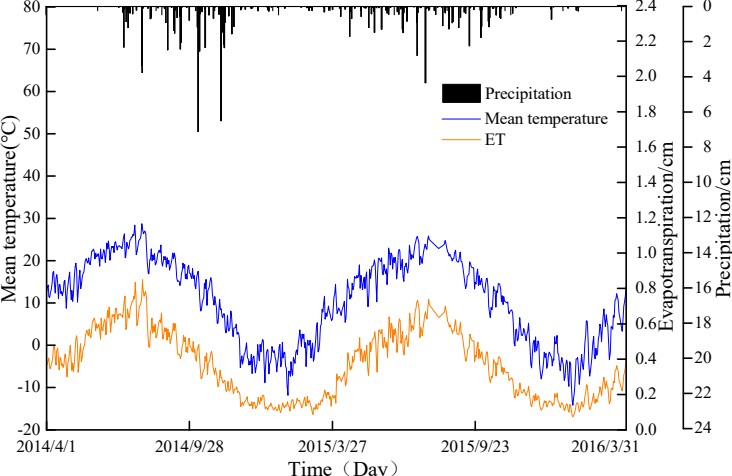

**Figure 2.** Daily precipitation, evapotranspiration, and mean temperature observed at the Ji county weather station from 1 April 2014, to 31 March 2016.

### 3.2. Soil Water Dynamics

The observed soil water dynamics in the typical soil layers are illustrated in Figure 3. The observed soil water contents in all of the soil profiles exhibited obvious seasonality during the two complete years from 1 April 2014 to 31 December 2016 (Figure 3). At the beginning of the rainy season, the antecedent average soil water content of the entire soil layer (0–200 cm) was only 14.01%. In the course of the rainy season, the soil water content exhibited a distinct increasing trend, especially during the continuous rainy periods.

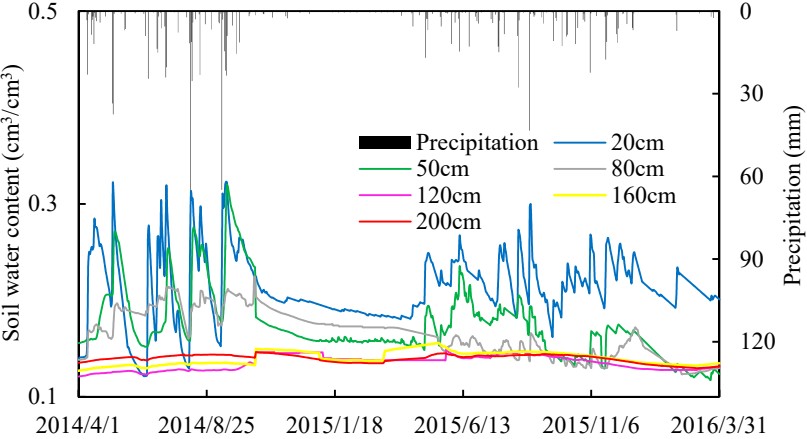

**Figure 3.** Temporal variations in the soil water dynamics at typical soil depths.

In addition, there were significant differences in the soil water content at different depths after rainfall. The soil surface moisture, especially the parts above 80 cm, was more sensitive to rainfall events with high amounts and intensities, and the soil water content ranged from 13.8% to 36.7%. During the observation period, there were three large-scale rainfall events in 2014, and the total precipitation was 71.1 mm, 123.6 mm, and 126.6 mm during these events. The soil water increments within 80 cm during the three rainfall events increased by 8.27%, 10.36%, and 13.24%, but the increase in the average soil water content between 100 cm and 200 cm was insignificant (less than 1%). The infiltration of the rainwater rapidly increased the water content of the surface soil, while the water content of the deep soil was low at that time. At the end of the rainy season, the soil water content decreased significantly within a few weeks. In particular, in the 10–50 cm soil layers, the soil water content decreased from 31.2% to 19.8% in 28 days. However, the soil moisture in the deep soil layers decreased at a very slow pace, and the soil dried out at a low rate throughout the dry season.

### 3.3. Optimization of the Model Parameters

We optimized the soil hydraulic parameters of the *Robinia pseudoacacia* forest using the Hydrus model. We used the measured dataset from 1 April 2013 to 31 October 2013 for the model calibration, and the dataset from 1 April 2016 to 31 October 2016 for the validation. The parameter optimization process aims to satisfy the convergence criterion, and it mainly optimized the empirical shape parameters: $n$, $\alpha$, and $K_s$. We used the calibrated hydraulic parameters (Table 2) to simulate the soil water dynamics from 1 April 2014 to 31 March 2016.

To analyze the accuracy of the HYDRUS-1D model, we compared the soil moisture observation data with the simulated data during the calibration period and the verification period (Figure 4). During the calibration period (Figure 4a), the data points tended to be distributed above the y = x line, indicating that the simulated water content was slightly higher than the measured water content; during the validation period (Figure 4b), the data tended to be distributed below the y = x line, indicating that the simulated value was slightly lower than the observed values. The evaluation index results for the calibration and validation periods showed that the $R^2$, *NSE*, *RMSE*, and *MAE* values of the calibration period were 0.79, 0.782, 0.035, and 0.064, respectively, and the $R^2$, *NSE*, *RMSE*, and *MAE*

values for the validation period were 0.85, 0.799, 0.028, and 0.035, respectively. The evaluation results for the verification period showed that the simulation process had a slight improvement and optimization. In most studies that used the Hydrus-1D to simulate the soil water dynamics in the Loess Plateau, the $R^2$ and *RMSE* values were mostly 0.7–0.84 and 0.015–0.063, respectively, when optimizing the hydraulic parameters [21,33]. When calibrating the hydraulic parameters in this study, the $R^2$ was 0.85, the *RMSE* was 0.028, and the evaluation index values were both within the range of the noted evaluation index results, indicating that the effectiveness of the model was acceptable, and the simulation values were in good agreement with the observed value. Thus, the optimized hydraulic parameters were appropriate for the simulation of soil water movement in the study area.

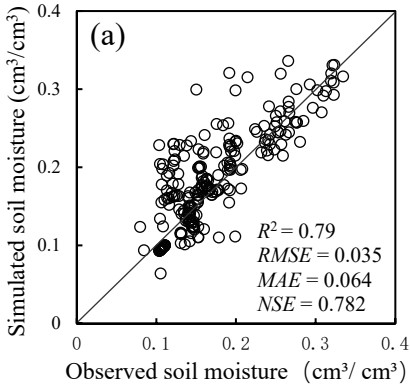 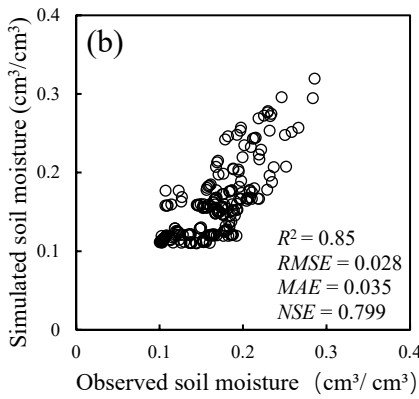

**Figure 4.** Comparison between the simulated and observed soil water contents: (**a**) the calibration period, and (**b**) the validation period, *n* = 252.

### 3.4. Simulation Accuracy of Soil Moisture Dynamics

We simulated the dynamic variations in the soil water content in the different soil layers of the forest using the HYDRUS-1D model and compared the simulated values and the measured soil water contents (Figure 5). It was evident that the dynamic variation in the simulated water contents of the soil profiles was consistent with the dynamic trend of the measured water contents. The range of the evaluation index results for the comparison between the simulated and measured water contents of each layer in the 0–200 cm soil profiles was as follows: $R^2$ of 0.7–0.93, *NSE* of 0.60–0.86, *RMSE* of 0.009–0.04, and *MAE* of 0.07–0.037. These results indicated that the simulated values for the different soil layers agreed well with the measured values. The average $R^2$, *NSE*, *RMSE*, and *MAE* values for 0–80 cm were 0.72, 0.75, 0.032, and 0.031, respectively, indicating that the goodness of fit between the simulated and measured values in the shallow soil was slightly lower. In the 160–200 cm soil layer, the $R^2$ and *NSE* values increased to 0.89 and 0.86, respectively, whereas both the *RMSE* and *MAE* decreased to 0.01, indicating that the model simulated the dynamics of the water content in the deep soil layer better. During the plant-growing season (May to September 2014), the average $R^2$, *NSE*, *RMSE*, and *MAE* values in the 0–200 cm soil profiles reached 0.7, 0.77, 0.03, and 0.03, respectively; in the non-growing season (December 2014 to March 2015), the average $R^2$, *NSE*, *RMSE*, and *MAE* values were 0.81, 0.84, 0.02, and 0.02, respectively. The evaluation results showed that the fitting precision in the plant-growing season was slightly lower than that in the non-growing season. In addition, we also found that during the precipitation period, the change in the simulated soil water contents of the 20 cm, 50 cm, and 80 cm soil layers had a hysteresis deviation compared with the measured soil water contents, Turkeltaub et al. [34] discovered that the simulated water content during the arrival of the wetting front is always higher than the measured water content, but this variation was observed only in the shallow soil layers.

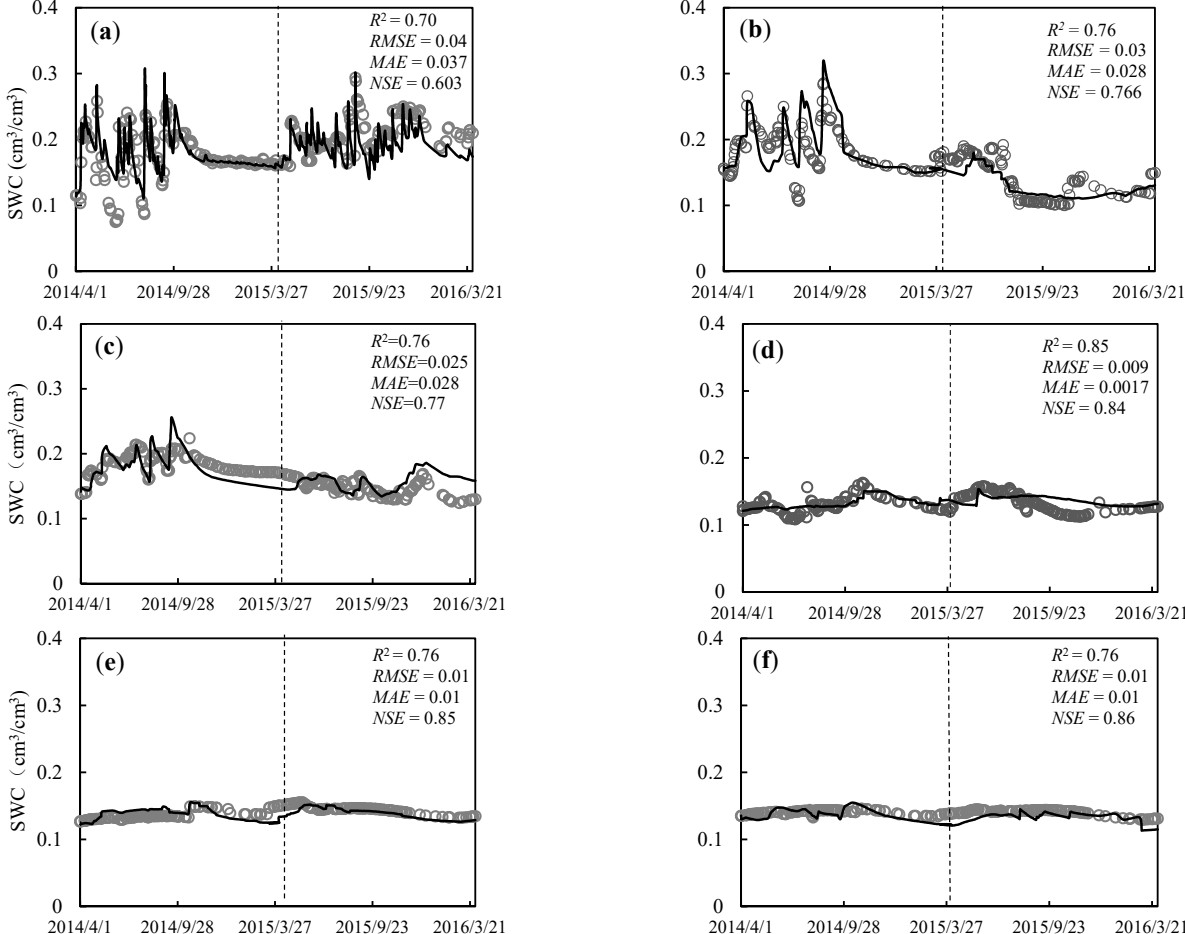

**Figure 5.** Comparison of the simulated and observed soil water contents at typical soil depths. The hollow circles represent the measured soil water contents, and the solid lines represent the simulated soil water contents. SWC is soil water content: (**a**) 20 cm, (**b**) 50 cm, (**c**) 80 cm, (**d**) 120 cm, (**e**) 160 cm, (**f**) 200 cm.

*3.5. Dynamics of the Water Flux and Estimation of Water Infiltration and Consumption*

The soil water flux is the amount of water passing through the soil layer per unit time, and the soil surface was taken as the reference plane in this study. The water flux was negative when the water infiltrated and moved downward during precipitation, and it was positive when the water moved upward during evaporation. The higher the absolute value of the water flux, the greater the amount of water passing through a particular soil layer. Two hundred and one simulation nodes were set at 1 cm intervals in the 0–200 cm soil layer in the model, and observation nodes (N ≤ 10) were set at 20 cm, 50 cm, 80 cm, 120 cm, 160 cm, and 200 cm, respectively, to simulate and calculate the water fluxes at the corresponding nodes. The responses of the water fluxes to rainfall in the different soil layers are plotted in Figure 6. The soil water flux in the shallow soil layer (0–80 cm) had a change pattern similar to that of precipitation, but there were differences in the degree of influence of the different rainfall amounts on the soil water flux. During the precipitation period, the values of the water fluxes of each soil layer gradually became negative from the surface layer to the deeper layers with the arrival of the wetting front, and the 0–80 cm layers were the most sensitive to precipitation. The water flux in the deep soil (160 cm and 200 cm) was relatively stable, and the annual average water flux was only 0.002 mm/d (negative value); however, it was replenished only when heavy rainfall occurred. The water flux in the deep soil is usually related to the cumulative amount of infiltration. After a long duration of heavy rainfall, the water flux of the deep soil gradually increased in the days after the rainfall ended. The time it took for the rainwater infiltration to travel from the surface layer to the 200 cm soil layer generally was greater than 10 days.

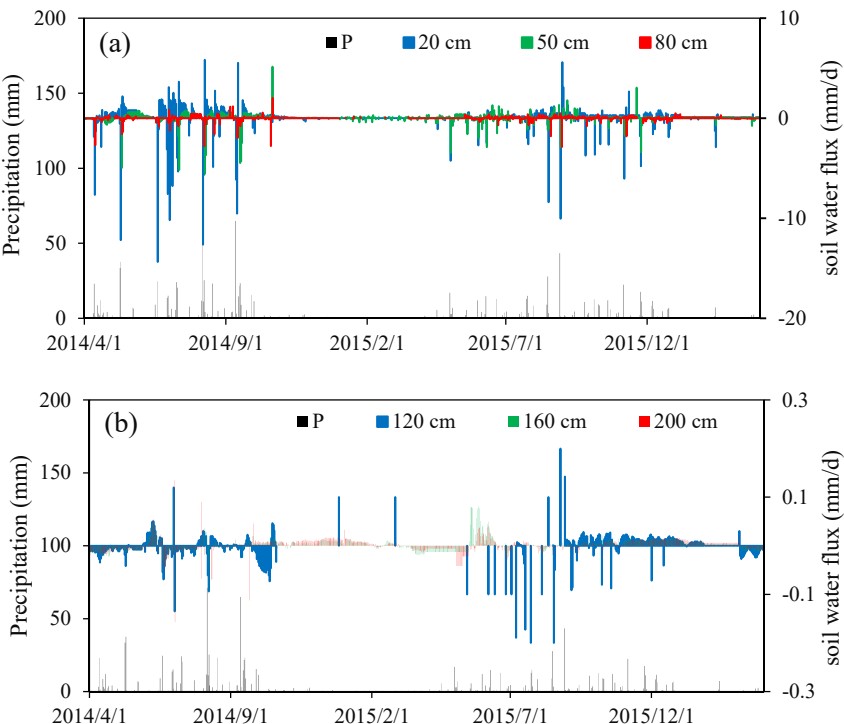

**Figure 6.** Changes in the soil water flux in the different soil layers and the precipitation distribution: (**a**) variations in the 20 cm, 50 cm, and 80 cm soil layer; and (**b**) variations in the 120 cm, 160 cm, and 200 cm soil layer. Note that the y-axis limits were adapted to the lower values in (**b**).

Table 3 shows a balanced situation of rainfall infiltration and soil water loss at the plot scale in the *Robinia pseudoacacia* forest during the simulation period. I and II represent hydrological years with precipitation of 652 mm and 455 mm, respectively (Table 3). During simulation period I, the upward and downward soil water fluxes within the 0–200 cm soil layer were 0.14 cm/d and 0.16 cm/d, respectively, that is, the evapo-transpiration and of the water that infiltrated into the soil were 524.6 mm and 616.8 mm, respectively. A total of 87.8% of the rainwater infiltrated into the soil and participated in the water cycle. The water surplus within the 0–200 cm soil layer in this hydrological year was calculated to be 92.2 mm, and the increase in the soil water storage was 64.05 mm. The percolation rate in the 200 cm soil layer was 0.077 mm/d, implying that 28.1 mm of this surplus infiltrated into the deeper soil horizons, and was potentially available for groundwater recharge.

**Table 3.** The water balance on the plot scale during the modelling period.

| Simulation Period | *P* (mm) | *I* (mm) | *E* (mm) | Δ*W* (mm) | *D* (mm) |
|---|---|---|---|---|---|
| I | 652 | 616.8 | 524.6 | 64.1 | 28.1 |
| II | 455 | 401 | 374.2 | 25.0 | 2.04 |

*Note: P* is precipitation, *I* is infiltration into the soil profile, *E* is the amount of evapo-transpiration, Δ*W* is the change in the soil water storage, and *D* is the amount of percolation at a depth of 200 cm.

During simulation period II, the upward and downward soil water fluxes within the 0–200 cm soil layer were 0.1 cm/d and 0.11 cm/d, respectively, that is, the water consumption and water infiltrated into the soil were 374 mm and 401 mm, respectively, so it can be calculated that the water surplus in the 0–200 cm soil layer in hydrological year I was 27 mm. The percolation rate at 200 cm was 0.0056 mm/d, which meant that only 2.04 mm of the soil surplus infiltrated into the deep soil water. Of this amount, 25 mm were used to increase the water storage and only 2 mm were used to recharge the water below 200 cm. A total of 90% of the rainwater that infiltrated into the soil was used for plant

transpiration and soil evaporation during the two hydrological years, which led to a small soil water recharge below 200 cm, but the water cycle within the 0–200 cm soil layer was generally in dynamic equilibrium. Note, however, that during years with below-average multi-year precipitation (579 mm), although the soil water storage in the 0–200 cm soil layer still increased, the increment corresponded to only about a 1% increase in the soil volumetric water content. Thus, the soil layers below 200 cm must be in a state of excessive depletion because the amount of water that infiltrated from the upper soil layers was limited, but the specific amount of depletion need to be further explored.

### 3.6. Simulation of Soil Water Movement under Different Rainfall Conditions

We found differences in the soil water flux changes under different rainfall conditions (Figure 6). To explore the effects of different rainfall conditions on soil water movement, we selected four different single rainfall events that occurred in summer and autumn to simulate the soil water dynamics before rainfall (W0), at the end of rainfall (W1), and 24 h after the end of the rainfall (W2). The amounts of precipitation during the four rainfall events were P1 (15 June 2014, light rain: 9.1 mm), P2 (19 June 2014, medium rain: 25 mm), P3 (6 August 2014, heavy rain: 71.1 mm), and P4 (12–15 September 2014, very heavy rain: 123.6 mm). The rainfall durations of the four rainfall events were 1.5 h, 7 h, 29 h, and 51 h, respectively. The simulation results showed that the increase in rainfall caused a direct increase in the infiltration depth, the amount of rainwater infiltration, and the increment of the soil water content. The depths and amounts of rainwater infiltration varied significantly at different soil depths after 24 h after the end of the rainfall (Figure 7). The four different rainfall amounts infiltrated to the 20 cm, 50 cm, 80 cm, and 140 cm soil layers, respectively, when the rainfall had just ended. In addition, the increment of the soil water content increased by 11.7%, 42.3%, 61.9%, and 68.7%, respectively. Twenty-four hours after the rainfall ended, the infiltration depth for the 9.1 mm and 25 mm rainfall events infiltrated to depths of 30 cm and 100 cm, respectively, indicating that light and medium rainfall caused an increase in soil water in the surface layer (0–30 cm) and the shallow layer (0–100 cm), respectively. In contrast, the infiltration depths of the 71.1 mm and 123.6 mm rainwater events reached depths of 160 cm and 200 cm, respectively. We found that only the P4 rainfall event caused a response in the soil water in the deep layer below 200 cm. Under the conditions of the four rainfall events with different amounts of precipitation, the amounts of rainwater infiltration were 8.08 mm, 19.30 mm, 65.19 mm, and 95.27 mm. The infiltration amounts of the four rainfalls accounted for the total rainfall in the order of P3 (91.82%), P1 (88.79%), P2 (77.2%), and P4 (77.08%).

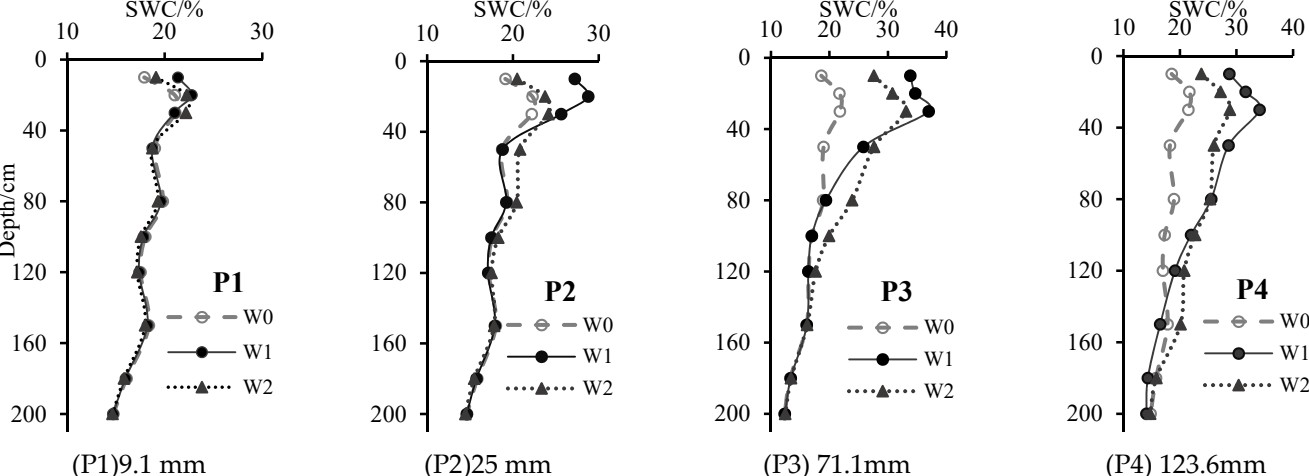

**Figure 7.** Variations in the soil water contents in the soil profiles before and after rainfall. W0, W1, and W2 are the soil water content before the rainfall, immediately after the rainfall, and 24 h after the end of the rainfall, respectively. SWC is the soil water content.

## 4. Discussion

### 4.1. Influence Analysis of the Model Parameters during the Calibration Process

The calibration of the model parameters is the key to determining the accuracy of the simulation results. We conducted these experiments to continuously calibrate and optimize the model's hydraulic parameters ($\alpha$, $n$, and $K_s$) by using the measured physical parameters, such as the soil grain size, soil bulk density, and field water holding rate, as the starting point for the adjustment of the parameters and to obtain a set of hydraulic parameters applicable to the unsaturated soil water movement at the sample site scale of a *Robinia pseudoacacia* forest in the loess area. Because of the spatial heterogeneity of the soil's microstructure, it was difficult to determine the soil's hydraulic parameters when extending from the sample plot scale to a larger scale. Variables such as rainfall and irrigation will not change the parameters of the van Genuchten-Mualem equation on which the model is based. This is because these parameters are determined by the soil properties such as the soil bulk density and soil particle size content. In the calibration process, we increased or decreased $\theta_r$ by 10% (while the other parameters remained unchanged), and it had little effect on the fitting effect of the simulated values, and the parameter sensitivity was low. The saturated hydraulic conductivity $K_s$ and shape parameter $n$ (Table 2) had greater impacts on the simulation results during the model calibration period, and the values of $K_s$ and $n$ predicted using the Rosetta module which is based on a neural network were large. The *RMSE* and *MAE* values decreased by 20% and 45% when $K_s$ and $n$ decreased to 0.5 times their original values. $\rho b$, $K_s$, and $n$ exhibit strong spatial correlations, that is, the deeper the soil layer was, the larger $\rho b$ and $n$ were; $K_s$ gradually became smaller as the depth of the soil layer increased. $\rho b$ and $n$ influenced $Ks$ through the pore-size distribution conditions when the soil texture was the same. Our results corroborated the results of Assouline [35] concerning the correlations between the soil bulk density and the hydraulic parameters in the vertical spatial distribution. Thus, the set of parameters obtained in this study should take into account the variability caused by the spatial heterogeneity of the soil when applied at larger scales, and this variability likely will occur for the shape parameter $n$ and the saturation hydraulic conductivity $K_s$.

### 4.2. Applicability Evaluation of the Model

We simulated the soil water movement in a *Robinia pseudoacacia* forest over two hydrological years. The evaluation results showed that the fitting results of the model were good. We found, however, that the simulated soil moisture line did not completely coincide with the observations in terms of the temporal dynamics, and there was hysteresis in the moisture simulation process, which has already been reported in previous studies [36,37]. We observed that the simulated values lagged behind the measured values in the 0–80 cm shallow soil layer (Figure 5), which was more obvious during the rainfall period. The main reasons for this may be that (1) the surface soil layer was greatly affected by meteorological factors, such as rainfall and evapotranspiration; and (2) the hydraulic parameters in each soil layer were fixed values that should not change with time. The simulation of the soil water movement during rainfall was relatively complicated, which led to the phenomenon in which the simulated values lagged behind the measured values in time. We concluded that the fitting effect was better for the deep soil moisture than for the shallow soil, which also confirmed that because of the influence of meteorological factors on the shallow soil during rainfall, the simulated value of the shallow soil water lagged behind the observed value. Precipitation was the only water source input in the model, and the impact of precipitation on the simulation accuracy was reflected in the temporal and spatial distributions. Precipitation and evapotranspiration may have affected the accuracy of the simulated soil moisture, especially for the surface soil, because it was the most sensitive to meteorological changes. Considering the influence of the uncertainties of the measured indicators, the actual sampling process relied on regular and fixed-point sampling, whereas the parameter values of the plants, soil, and atmospheric factors required by the model were continuous data. Therefore, the accuracy of the leaf area index measurements [38],



the sample size of the root distribution [39], and the meteorological conditions during a long-period series [40,41] inevitably affected the coupling results of each module and finally were reflected in the soil water movement simulation results. This may be the reason for the remaining discrepancy between the simulated and observed data. On the basis of this analysis, we could improve the deviation in the boundary conditions, root water absorption, and other modules by increasing the sampling points and sample size within the spatiotemporal range [42].

*4.3. Spatiotemporal Dynamic Estimation Analysis of Rainfall, Soil Moisture and Water Balance*

In the process of simulating soil water movement before and after rainfall events, we found that the rainfall amount had a significant influence on rainwater infiltration, soil moisture, and its redistribution. Several studies conducted on the Loess Plateau have shown that the position of the wetting front is almost 120 cm under natural rainfall conditions [43]. Our results showed that the average depth of rainwater infiltration was around 100 cm under moderate rainfall, whereas the depth of precipitation infiltration could be more than 200 cm under heavy rainfall. Thus, the infiltration depth of the rainwater and the amount of infiltration were basically positively correlated with the amount of rainfall. Due to the limited penetration depth of the rainfall in the soil, the deep soil moisture remained in a stable state, and it was mainly supplied by the rainwater from heavy rainfall events with a long duration [44,45]. In the 71 mm and 123.6 mm rainfall events, the rainfall durations reached 29 h and 51 h, respectively (Figure 3). The rainwater that infiltrated into the soil caused the soil water content to rapidly increase, and the water potential was at a high level at this time, but the water potential of the deep soil was very low. The difference in the water potential caused the wetting front in the soil to continuously move into the lower soil. The wetting front gradually decreased with the disappearance of the water potential difference until it stopped. Please note that in our experiments, we selected single independent rainfall events, whereas the natural conditions in the loess area included intermittent continuous rainfall, and thus the rainfall and infiltration characteristics were more complicated. This part of the study will require long-term observations and analysis to fully describe the relationship between soil water movement and rainfall in the future.

From the perspective of the water balance in the *Robinia pseudoacacia* forest, the water balance at the forestland scale is dynamic. Previous studies have reported the water balance budget under different land use conditions, especially in farmland [46,47], and they have shown that the supply of precipitation water is compensated by evaporation in farmland. Our results, however, showed that the rainwater that infiltrated into the soil was used primarily for evapotranspiration, especially transpiration. The infiltration and consumption activities indicated that the water cycle in the loess area mainly involved vertical water exchange. The soil layers above 120 cm were the main active layers for rainwater infiltration and soil water flux movement, which was the main distribution area of the *Robinia pseudoacacia* root system (Figure 6). The transpiration of the *Robinia pseudoacacia* accounted for about 70% of the rainfall during the simulation period (Table 3). The vigorous transpiration in the soil root zone reflected the root distribution and water absorption of the *Robinia pseudoacacia*. In hydrological years in which the rainfall was much lower than the perennial average rainfall, the soil water storage was significantly reduced. The percolation rate in the 200 cm soil layer was only 0.0056 mm/d, with almost no water recharge to the deep soil. In the dry year (II), the water flux in the surface layer was 22 times (Figure 6) that in the deep layer (200 cm). Light rain and moderate rain could not recharge the water in the deep soil. The deep soil layer received precipitation only under heavy rainfall to supply water absorption to the deep roots of the *Robinia pseudoacacia* [48,49].

Ensuring the increase in deep soil water storage is the key to the growth of plants in this area. In addition to rainfall, the other factors affecting the deep soil water content include the soil particle size, *Ks* and the value of the shape parameter *n*. The increase in the clay content of the deep soil (Table 2) led to a subsequent decrease in the water flux at the bottom, which affected the replenishment of the deep soil moisture. Similar results have

been obtained in previous studies [50,51]. In addition, many researchers have reported on deep leakage estimation. Min et al. [52] simulated and predicted a deep soil leakage rate of 223 mm/y for the North China Plain. Ries [53] estimated that the percolation rate in the semi-arid unsaturated soil area was at least 310 mm/y. Our study showed that the annual average percolation rate was only 2.04 mm/y in the dry hydrological year. Because Min et al.'s research was carried out in farmland, the infiltration rate may have been higher as a result of human factors (reclamation and irrigation measures). In this study, the soil in the *Robinia pseudoacacia* forest land had not been disturbed by reclamation, and the actual migration rate of the wetting front was only 0.17 mm/d because of the high soil density and low permeability. Although many previous studies have investigated the effects of precipitation on the soil water redistribution process under dry conditions [54,55], studies of extreme rainfall events are lacking. The present study revealed that the low soil water content mainly occurred in the lower layer of the root distribution zone where the roots have difficulty absorbing water. If the annual rainfall continues to decrease, the deep soil moisture cannot be replenished, which will cause the deep roots to use the deeper soil water. This situation is not conducive to the growth of vegetation. To effectively solve this problem, measures such as adjusting the forest density or land preparation methods (level terrace) can be considered to improve the rainfall infiltration into the soil profile and the soil water storage during actual afforestation [56].

## 5. Conclusions

We simulated the spatiotemporal movement of soil water in a *Robinia pseudoacacia* forest established by the GFGP in the loess area using the HYDRUS-1D model. We also investigated transport and replenishment of the soil water in the root zone under different rainfall events. The model with the calibrated parameters described the soil water movement process in the 200 cm root zone of the forest better. In this model, the growth season and the deep soil had a better fit in terms of the water movement dynamics. Precipitation, root distribution characteristics, and soil texture were the main factors that affected the accuracy of the simulation. In addition, the numerical simulation of water flow under different rainfall events enabled us to understand the main infiltration mechanism in the forestland in the loess area. The interannual variation in the soil water flux was significant, and it was strongly dependent on the soil depth, time distribution, and seasonal rainfall. The depth and amount of rainwater infiltration were proportional to the amount of rainfall. Furthermore, the response of the deep soil water to rainfall infiltration exhibited a lag effect when the water infiltrated through the entire soil profile (0–200 cm), and the lag period was at least 10 days. The active layer of the rainwater infiltration and the soil water flux transfer in the study area was distributed mainly in the soil layer above 120 cm, which effectively promoted the absorption of water by plant roots. However, when the annual rainfall was less than 455 mm, the deep soil of a *Robinia pseudoacacia* forest may suffer from water shortages, especially from June to September, causing the root system to use the deeper soil water, and leading to the development of dry soil layers. Four hundred and fifty-five millimeters may be the limit for the formation of a dry soil layer.

The results of this study improve our understanding of the infiltration mechanism of one-dimensional water flow in unsaturated soils, and provide a basis for evaluating the characteristics of water infiltration and wetting front migration in the loess area. However, this research did not consider two aspects: (1) the soil water movement in the soil layer below 200 cm; and (2) whether the time scale during our research could influence the accuracy and representativeness of the research results. Therefore, our future research the percolation below 200cm will be tracked, and the time scale of the study will be increased.

**Author Contributions:** Conceptualization, Y.L. and J.Z.; Data curation, Y.L., R.S. and M.S.; Formal analysis, Y.L.; Funding acquisition, J.Z.; Investigation, R.S. and M.S.; Methodology, Y.L. and J.Z.; Project administration, J.Z.; Resources, J.Z.; Software, Y.L.; Supervision, J.Z. and Y.Y.; Validation, Y.L.; Visualization, Y.L.; Writing—original draft, Y.L.; Writing—review & editing, Y.Y. and J.Z. All authors have read and agreed to the published version of the manuscript.

**Funding:** This research was supported by the National Key Research and Development Program of China (No. 2016YFC0501704).

**Acknowledgments:** We would like to thank Rongwei Zhao, Baoqiang Chen and Yaqiong Wang for their hospitality and support during fieldwork.

**Conflicts of Interest:** The authors declare no conflict of interest.

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
