# Peer review of "Simulation of Soil Water Dynamics in a Black Locust Plantation on the Loess Plateau, Western Shanxi Province, China"

_water, doi:10.3390/w13091213_

Round 1

Reviewer 1 Report

The paper tries to simulate the soil water dynamics in a black locust plantation in China. The topic is appropriate and interesting for the Water journal. The title and abstract are concise and clearly describe the contents of the paper. The presentation is clear and the language is fluent. Methods are clearly described and can be applied by other scientists. However, in my opinion, in its present form, the paper needs some clarifications before being published in this journal. The main aspect to be reviewed is the following.

- What is the objective of your study? I mean, what is the scientific question to be addressed? Currently, I can only see a mere exercise of modeling which, of course, is well addressed and performs the quality requirements. However, I don't understand why this study is important and should be published. What novelty does your study bring? If there is no new to say, the paper must not be published. This assertion is also supported by the Conclusion chapter, where there is nothing new that is not currently known. Please, refocus your study to say something new. It must clearly be visible and identified on the document.

- I believe the study has a high potential because ecosystems from this location are infrequent in literature. In this present form, the study represents a very modest contribution to our understanding, but I think that, if authors carry out a better organizational focus on the main ideas throughout the text and rewrite the study, it would help the article a lot and it would be very close to being an important contribution. I think the study hides more work than the one is presented. I hope that the enclosed comments will assist to authors in revising it for a possible publication.

Author Response

Revision Note (Water-1411911)

We gratefully thank the editors and reviewers for their time and their insightful comments and suggestions, which have significantly increased the quality of the manuscript and have enabled us to improve the manuscript. Point by point responses to the comments of the reviewers and descriptions of the revisions are provided below.

Comments on manuscript “Simulation of soil water dynamics in a black locust plantation on the Loess Plateau, Western Shanxi Province, China”

Comment 1: The paper tries to simulate the soil water dynamics in a black locust plantation in China. The topic is appropriate and interesting for the Water journal. The title and abstract are concise and clearly describe the contents of the paper. The presentation is clear and the language is fluent. Methods are clearly described and can be applied by other scientists. However, in my opinion, in its present form, the paper needs some clarifications before being published in this journal.

Response 1: Thank you very much for your positive comments on our manuscript. We have addressed all of your comments in the revised manuscript. We also corrected the grammar and spelling errors. Furthermore, the point-by-point responses to each reviewers’ comments are provided below, along with a clear indication of the location of the revisions. We hope these changes will make our manuscript acceptable for publication.

Comment 2: What is the objective of your study? I mean, what is the scientific question to be addressed? Currently, I can only see a mere exercise of modeling which, of course, is well addressed and performs the quality requirements. However, I don't understand why this study is important and should be published. What novelty does your study bring? If there is no new to say, the paper must not be published. This assertion is also supported by the Conclusion chapter, where there is nothing new that is not currently known. Please, refocus your study to say something new. It must clearly be visible and identified on the document.

Response 2: We greatly appreciate your comments. We re-examined our article and found that the research objectives were indeed not clearly stated. We revised the statement of the objectives of our study. See lines 51-55, 59-61 and 109-112.

Comment 3: I believe the study has a high potential because ecosystems from this location are infrequent in literature. In this present form, the study represents a very modest contribution to our understanding, but I think that, if authors carry out a better organizational focus on the main ideas throughout the text and rewrite the study, it would help the article a lot and it would be very close to being an important contribution. I think the study hides more work than the one is presented. I hope that the enclosed comments will assist to authors in revising it for a possible publication.

Response 3: Thank you for your valuable and positive comments. We have reorganized the main ideas throughout the text to make the expressions more complete. The revised parts include the introduction, materials and methods, and discussion sections.

In summary, we tried our best to improve the manuscript and made several changes to the manuscript. These changes do not influence the main content and framework of the paper. We did not list the changes, but we did highlight them in blue in the revised paper.

We earnestly appreciate the Editors/Reviewers’ helpful and positive work, and we hope that the corrections will meet with their approval. We would be glad to respond to any further questions and comments that you may have.

Reviewer 2 Report

The paper describes the water fluxed and soil water profiles over two years for a forest planted with black locust. Water contents measured at several depths (every 10 cm over the first meter and every 20 cm till 2 meters) were modeled using Hydrus 1D model to resolve Richards’ equations. The hydraulic parameters were fitted for each layer according to the soil texture and fitting the water content measures. The modeling is quite concluding, and conclusions can be stated on the water budget at the studied soil profile scale (0-200 cm). The results are interesting and adequately discussed. They show the importance of precipitation to recharge water moisture over the soil profile and below. 

However, there are several significant improvements to be done before publication. The main points to address are the following:

  • The authors need to properly introduce their work within the framework of previous studies and state of art. The scientific question is not stated at all. The authors present their research object (the water budget in the site) but do not deal with any scientific question. These scientific questions could be as follows. Do dry years inhibit moisture recharge at deep horizons? Or how does the water stock evolve between years with regards to soil plants’ need for water? Etc.
  • The modeling approach is poorly presented:
    • I know that the modeled results are probably good since Hydrus 1D resoled the good equations, but there are many typos in equations. Also, some correspond to a vertical axis oriented upward, and others to the z-axis oriented downward. Several symbols vary in the text and between figures.
    • No pieces of information are given for the mesh and the time discretization. The numerical domain and initial and boundary conditions should be stated clearly.
    • The strategy of inversion with the list of parameters that are fixed and those that are optimized should be clearly stated.
  • Some points and results are weird and deserve more explanations. I have highlighted several parts in the text that need to be improved. The color of textboxes indicates the degree of importance of the comments: green (= comment), yellow (watch = must be checked), orange (advisory = must be addressed), and red (warning = must be addressed with attention). 

Great efforts are needed to improve the manuscript. I have enclosed a document with all my suggestions. I let the authors improve their work. I encourage them to submit a revised version that I will be happy to review.

Author Response

Revision Note (Water-1411911)

We gratefully thank the editors and reviewers for their time and their insightful comments and suggestions, which have significantly increased the quality of the manuscript and have enabled us to improve the manuscript. Point by point responses to the comments of the reviewers and descriptions of the revisions are provided below.

Comments and suggestions for Authors

The paper describes the water fluxed and soil water profiles over two years for a forest planted with black locust. Water contents measured at several depths (every 10 cm over the first meter and every 20 cm till 2 meters) were modeled using Hydrus 1D model to resolve Richards’ equations. The hydraulic parameters were fitted for each layer according to the soil texture and fitting the water content measures. The modeling is quite concluding, and conclusions can be stated on the water budget at the studied soil profile scale (0-200 cm). The results are interesting and adequately discussed. They show the importance of precipitation to recharge water moisture over the soil profile and below.

Response: Thank you very much for taking the time to review our manuscript, We greatly appreciate all of your positive comments and suggestions on our manuscript! We have addressed all of your comments in the revised manuscript. Please find my itemized responses below and my revisions in the re-submitted files. In addition, since the text has been modified throughout the manuscript, we used yellow and blue to highlight the modified text. Thus, there is no special motation of the modified line and page number.

Comment 1: The authors need to properly introduce their work within the framework of previous studies and state of art. The scientific question is not stated at all. The authors present their research object (the water budget in the site) but do not deal with any scientific question. These scientific questions could be as follows. Do dry years inhibit moisture recharge at deep horizons? Or how does the water stock evolve between years with regards to soil plants’ need for water? Etc.

Response 1: We greatly appreciate your comment on the objectives of the study. It is in fact true that they were not clearly stated in the previous version. Regarding the comment about the objectives of the study and the work conducted in previous studies, we have addressed and added this information in the revised manuscript.

Comment 2: I know that the modeled results are probably good since Hydrus 1D resoled the good equations, but there are many typos in equations. Also, some correspond to a vertical axis oriented upward, and others to the z-axis oriented downward. Several symbols vary in the text and between figures.

Response 2: Thank you for your professional and detailed comments and suggestions. It is very important to standardize diagrams and equations. The incorrect parts of the equations and diagrams have been modified according to your comments that highlighted areas in the previous manuscript. In addition, we have improved several parts of the text that you highlighted. Thank you very much for correcting some of the terms in the manuscript, and for explaining some of the hydrological phenomena, which greatly improved the quality of the revision of the manuscript.

Comment 3: No pieces of information are given for the mesh and the time discretization. The numerical domain and initial and boundary conditions should be stated clearly.

Response 3: Thank you very much for pointing out the lack of this content. We added the information regarding the time discretization, numerical domain and initial and boundary conditions in the materials and methods section.

Comment 4: The strategy of inversion with the list of parameters that are fixed and those that are optimized should be clearly stated.

Response 4: Thank you very much for your comments. We have revised the part of concerning the optimization of the parameters and have emphasized the parameters that needed to be fitted or fixed.

Comment 5: Some points and results are weird and deserve more explanations. I have highlighted several parts in the text that need to be improved. The color of textboxes indicates the degree of importance of the comments: green (= comment), yellow (watch = must be checked), orange (advisory = must be addressed), and red (warning = must be addressed with attention).

Response 5: We improved the manuscript according to your comments. The revised content in the manuscript is highlighted in yellow and blue. In addition, we modified some of the content of the results and discussion sections and added some missing results information that needed to be explained. Thank you again for all your comments and suggestions. They were very helpful in revising of the manuscript.

In summary, we tried our best to improve the manuscript and made several changes to the manuscript. These changes do not influence the main content and framework of the paper. We did not list the changes, but we did highlight them in blue in the revised paper.

We earnestly appreciate the Editors/Reviewers’ helpful and positive work, and we hope that the corrections will meet with their approval. We would be glad to respond to any further questions and comments that you may have.

Reviewer 3 Report

Dear Authors;

I found your paper to be interesting and useful to the field of soil science, hydrology, and ecohydrology.  I appreciate very much your efforts to model water flow in unsaturated soils and believe that your efforts will improve our understanding of how water moves through the soil. 

I suggest that you give more detail in the introduction about the differences between saturated and unsaturated soil water flow.  This paper is extremely technical and difficult to follow for readers interested in soil water flow dynamics but it is not presented in a way that will be attractive to many readers who simply want to know why it is important to understand unsaturated flow and how those differences apply in a much broader area than the loess plateau in China.  A lot of the mathematical modeling could perhaps be moved to supplemental information for those scientists who want to know the model specifics.  This would shorten your paper and make it easier to read and more attractive to a much broader audience.  Therefore, you should better describe the knowledge gap and inform the reader what the actual problem is that you are trying to solve with your research.  I think that making these corrections would  would help to improve the paper a lot.    

Some minor edits:

Line 47 does not make sense:  "The Loess Plateau is the most concentrated loess area of loess in the world, and it is 47
located in the northern part of China"

Line 52: add what the annual precipitation is.

Line 53:  this sentence is written wrong and is hard to understand "Thus, whether the vegetation ecosystem 53
formed by the GFGP exceeds the soil water carrying capacity has become a research 54
hotspot in academia and an urgent scientific problem that needs to be solved."

Line 60:  Can you give more detail about the soil structure here?

Line 62:  you use the word water "carrying" capacity.  Please change throughout your paper to read "holding" capacity.

Line 70 - 71:  This might be a good place to describe the differences between unsaturated flow and saturated flow and to give the reader an understanding of why simulations are needed.  You need to better describe the the exact problem is that you are trying to solve or what knowledge gap you are trying to fill.

Line 80:  please describe in more detail what constant and non constant boundary conditions are.

Line 165:  please describe what "standard wood" is

Line 175-177:  There should be a citation associated with this sentence "The Hydrus-1D model is based on Richard’s equations and uses the Galerkin linear
finite element method to spatially discretize the soil profile while allowing for time-variable boundary conditions."

Line 191:  please again describe what the boundary conditions are in more detail.  

Line 237:  please change "are is" to "is"

Line 258:  Is 14.01% the average of all of the depths measured?

Line 268:  This result "between 1 m and 2 m was only 0.78%" does not appear to match the graph.

Line 269:  why do you keep changing your units between mm to cm to m.  Please try to be consistent throughout with your units.

Figures 3 and 6 are difficult to read and differentiate the lines from one another.

In the discussion you should give more detail about how the model parameters are adjusted for the change between unsaturated flow to saturated flow as rain begins to infiltrate the soil and percolate through the soil.

Author Response

Revision Note (Water-1411911)

We gratefully thank the editors and reviewers for their time and their insightful comments and suggestions, which have significantly increased the quality of the manuscript and have enabled us to improve the manuscript. Point by point responses to the comments of the reviewers and descriptions of the revisions are provided below.

Comments and suggestions for Authors

I found your paper to be interesting and useful to the field of soil science, hydrology, and ecohydrology.  I appreciate very much your efforts to model water flow in unsaturated soils and believe that your efforts will improve our understanding of how water moves through the soil.

I suggest that you give more detail in the introduction about the differences between saturated and unsaturated soil water flow.  This paper is extremely technical and difficult to follow for readers interested in soil water flow dynamics but it is not presented in a way that will be attractive to many readers who simply want to know why it is important to understand unsaturated flow and how those differences apply in a much broader area than the loess plateau in China. A lot of the mathematical modeling could perhaps be moved to supplemental information for those scientists who want to know the model specifics.  This would shorten your paper and make it easier to read and more attractive to a much broader audience.  Therefore, you should better describe the knowledge gap and inform the reader what the actual problem is that you are trying to solve with your research. I think that making these corrections would help to improve the paper a lot.

Response: Thank you for the positive and very interesting comments on our manuscript. It is important to clearly state the objectives of the research. We have addressed all of your comments and suggestions in the revised manuscript. In addition, based on your comments and the comments from the other reviewers, the research objectives now clearly stated. Our responses to your individual comments are provided below.

Comment 1: Line 47 does not make sense: "The Loess Plateau is the most concentrated loess area of loess in the world, and it is 47 located in the northern part of China"

Response 1: Thank you very much for your comments. Our statement is indeed not objective enough and may misguide readers. We have modified the description of the ecological status of the study area (Lines 48-49, Page 2).

Comment 2: Line 52: add what the annual precipitation is.

Response 2: We have added the annual precipitation in the revised manuscript in parentheses (Line 58, Page 2).

Comment 3: Line 53: this sentence is written wrong and is hard to understand "Thus, whether the vegetation ecosystem 53

formed by the GFGP exceeds the soil water carrying capacity has become a research 54

hotspot in academia and an urgent scientific problem that needs to be solved."

Response 3: Thank you for your comments. We have modified the inappropriate expression in the revised manuscript (Lines 59-62, Page 2).

Comment 4: Line 60: Can you give more detail about the soil structure here?

Response 4: Thank you very much for your suggestions. We have added the description of the complexity of the soil structure in the revised manuscript (Lines 66-68, Page 2).

Comment 5: Line 62: you use the word water "carrying" capacity.  Please change throughout your paper to read "holding" capacity.

Response 5: Thank you very much for your comments. We have changed the expression “water carrying capacity”, which may misguide readers (Line 69-71, Page 2).

Comment 6: Line 70 - 71: This might be a good place to describe the differences between unsaturated flow and saturated flow and to give the reader an understanding of why simulations are needed.  You need to better describe the the exact problem is that you are trying to solve or what knowledge gap you are trying to fill.

Response 6: Thank you very much for your suggestions. We have described the differences between unsaturated flow and saturated flow (Lines 72-79, Page 2), and have added the research objectives (Lines 111-114, Page 3).

Comment 7: Line 80: please describe in more detail what constant and non-constant boundary conditions are.

Response 7: Thank you very much for your comments. We have added the boundary conditions and have described which conditions are constant or non-constant in parentheses in the revised manuscript (Line 95-97, Page 2).

Comment 8: Line 165: please describe what "standard wood" is

Response 8: Thank you very much for your comments. We have replaced the vague expression of “standard wood” with “representative wood with an average tree height and crown width” in the revised manuscript (Line 200, Page 3).

Comment 9: Line 175-177: There should be a citation associated with this sentence "The Hydrus-1D model is based on Richard’s equations and uses the Galerkin linear

finite element method to spatially discretize the soil profile while allowing for time-variable boundary conditions."

Response 9: Thank you very much for your suggestions. We have added a reference to the sentence "The HYDRUS-1D model is based on Richard’s equations and uses the Galerkin linear finite element method to spatially discretize the soil profile while allowing for time-variable boundary conditions [28]" (Line 212, Page 6).

Comment 10: Line 191: please again describe what the boundary conditions are in more detail.

Response 10: Thank you very much for your comments. We have revised this part of the description based on your comments and those of the other reviewers. We now describe the initial conditions, the upper boundary conditions, and the lower boundary conditions in the section concerning the initial conditions and boundary conditions (Lines 223–231, Page 6).

Comment 11: Line 237: please change "are is" to "is"

Response 11: Thank you very much for your comments. We have changed “are is” to “is” in the revised manuscript (Line 286, Page 9).

Comment 12: Line 258: Is 14.01% the average of all of the depths measured?

Response 12: Thank you very much for your comments. We have modified this vague phrase to “the antecedent average soil water content of the entire soil layer (0–200 cm) was only 14.01%” (Lines 307–310-311, Page 9).

Comment 13: Line 268: This result "between 1 m and 2 m was only 0.78%" does not appear to match the graph.

Response 13: Thank you very much for your comments. We have modified the inappropriate expression in the revised manuscript (Lines 318–320, Page 9).

Comment 14: Line 269: why do you keep changing your units between mm to cm to m. Please try to be consistent throughout with your units.

Response 14: We appreciated your comments on the expression problem. We have unified the expression of the units in the revised manuscript (Full text revised).

Comment 15: Figures 3 and 6 are difficult to read and differentiate the lines from one another.

Response 15: Thank you very much for your comments. We have adjusted the line spacing, width, and color in Figures 3 and 6 to make it easier to read. We have also adjusted the format of the horizontal and vertical coordinates.

Comment 16: In the discussion you should give more detail about how the model parameters are adjusted for the change between unsaturated flow to saturated flow as rain begins to infiltrate the soil and percolate through the soil.

Response 16: Thank you very much for your comments. I am sorry that this section was not clear in the original manuscript. It is a very important issue that may misguide readers and really needed to be explained in the manuscript. In fact, we adjusted the hydraulic parameters (θr, n, and Ks) by fixing the soil particle size, soil bulk density, and field water holding capacity of each soil layer. The hydraulic parameters, including α, l, and m are mainly affected by the soil properties (soil particle size and soil bulk density), which were measured in the experimental plots and laboratory. The input variables such as the rainfall and irrigation do not change the parameters of the van Genuchten-Mualem equation on which the model is based. In the period of calibration, we found that the hydraulic parameters Ks and n were more sensitive when we fixed the other parameters. Then, we focused on the calibration of Ks and n. According to your comments, we added this content to the discussion of how we calibrated the parameters that are sensitive and insensitive and hope that this will be more suitable and will improve the manuscript (Lines 492–498, Page 15).

In summary, we tried our best to improve the manuscript and made several changes to the manuscript. These changes do not influence the main content and framework of the paper. We did not list the changes, but we did highlight them in blue in the revised paper.

We earnestly appreciate the Editors/Reviewers’ helpful and positive work, and we hope that the corrections will meet with their approval. We would be glad to respond to any further questions and comments that you may have.

Round 2

Reviewer 1 Report

I think the paper has been improved from the suggestions.